# Survival and Physiological Recovery after Capture by Hookline: The Case Study of the Blackspot Seabream (*Pagellus bogaraveo*)

Ignacio Ruiz-Jarabo [1,2,*], Miriam Fernández-Castro [1], Ismael Jerez-Cepa [1], Cristina Barragán-Méndez [1], Montse Pérez [3], Evaristo Pérez [3], Juan Gil [4], Jesús Canoura [5], Carlos Farias [4], Juan Miguel Mancera [1,†] and Ignacio Sobrino [4,†]

[1]   Department of Biology, Faculty of Marine and Environmental Sciences, Universidad de Cádiz, Campus de Excelencia Internacional del Mar (CEI-MAR), Av. República Saharaui s/n, Puerto Real, E-11510 Cádiz, Spain; miriam.fernandez@uca.es (M.F.-C.); ismael.jerez@uca.es (I.J.-C.); cristina.barragan@uca.es (C.B.-M.); juanmiguel.mancera@uca.es (J.M.M.)
[2]   Department of Animal Physiology, Faculty of Biological Sciences, University Complutense, C. José Antonio Novais, 12, Ciudad Universitaria, E-28040 Madrid, Spain
[3]   Instituto Español de Oceanografía (IEO-CSIC), Centro Oceanográfico de Vigo, E-36390 Vigo, Spain; montse.perez@ieo.es (M.P.); evaristoperezr@hotmail.com (E.P.)
[4]   Instituto Español de Oceanografía (IEO-CSIC), Centro Oceanográfico de Cádiz, Muelle de Levante, s/n, Puerto Pesquero, E-11006 Cádiz, Spain; juan.gil@ieo.es (J.G.); carlos.farias@ieo.es (C.F.); ignacio.sobrino@ieo.es (I.S.)
[5]   TRAGSATEC, C/Julián Camarillo, 6B, E-28037 Madrid, Spain; jesus.canoura@ieo.es
*     Correspondence: ignaru02@ucm.es
†     These authors contributed equally to this study.

**Abstract:** Evaluating the survival of discarded species is gaining momentum after the new European Common Fisheries Policy (Article 15 of the European Regulation No. 1380/2013). This regulation introduced a discard ban, with an exemption for those species with demonstrated high survival rates after their capture and release. Candidate species should be evaluated for each fishing gear and geographical area. In this study, we assessed not only survival, but also physiological recovery rates of blackspot seabream (*Pagellus bogaraveo*) below commercial size captured with a hookline called "*voracera*" in the Strait of Gibraltar (SW Atlantic waters of Europe). Experiments onboard a commercial fishing vessel were paralleled with studies in controlled ground-based facilities, where the capture process was mimicked, and physiological recovery markers were described. Our results confirmed that hookline capture induced acute stress responses in the target species, such as changes in plasma cortisol, lactate, glucose, and osmolality. However, 90.6% of the blackspot seabreams below commercial size captured with this fishing gear managed to survive, and evidenced physiological recovery responses 5 h after capture, with complete homeostatic recovery occurring within the first 24 h. Based on this study, the European Commission approved an exemption from the discard (EU Commission Delegated Regulation 6794/2018). Thus, the robust methodology described herein can be an important tool to mitigate the problem of discards in Europe.

**Keywords:** capture-recapture; discards; fisheries policy; physiology; survival

## 1. Introduction

Nowadays, under the new Common Fisheries Policy and according to the Article 15 of the European Regulation (EU) N° 1380/2013, discards should be introduced as landings. This European compulsory landing obligation affects all captured species that are subject to catch limits and, in the Mediterranean Sea, also catches of species that are subject to minimum sizes. However, as reported in Article 13 of the regulation, captured animals could be released back into the sea if robust scientific evidence indicates high survival rates.

Survival of captures depends on many factors including the fishing gear employed [1], the captured species [2], and environmental variables [3] amongst others [4,5]. Thus,

survival of discards should be evaluated for each fishing gear and geographic area [6]. There are many studies evaluating the vitality of captured fish as a golden method to infer their survival capacity, which are based on scoring the impairment of reflexes or the degree of external injuries [7]. However, it was described that fish released with no obvious signs of injury often show delayed mortalities days after being released [8,9]. In this sense, it was described that fishing processes elicited acute stress responses in captured fish [10,11], which caused physiological imbalances that could last for a longer time.

Once the organism detects some stimulus (external or internal) that may pose a danger or situation susceptible to disturb the basal physiological balance, a stressful situation occurs. Then, a series of responses are triggered, which are categorized as primary, secondary, and tertiary responses [12,13]. Primary responses in teleost fish include the release of catecholamines (adrenaline and noradrenaline) from chromaffin tissue and cortisol (as the main corticosteroid hormone in teleosts) from the hypothalamus–pituitary–interrenal axis [14]. In species from different taxa (*Sparus aurata*, *Solea senegalensis*, and *Colossoma macropomum*), cortisol reached its maximum concentrations in blood within the first 60 min after an acute stress situation [15–17], recovering its basal levels between 4 h and 24 h after the challenge [18], depending on the intensity of the stimuli, species, and environmental conditions. Secondary responses are promoted by the action of these hormones, making oxygen and metabolic substrates available to demanding tissues [19]. The analysis of plasma just after capture revealed the onset of anaerobic glycolysis coupled to the production of lactate [20]. Energy stores are thus mobilized, and the intermediary metabolism is modified along with hydromineral imbalances related to changes in plasma osmolality in marine fish [21]. Therefore, allostatic changes are necessary to regain balance [22]. If the stressful situation extends over time and/or a new homeostatic state is not recovered, the energy reserves may be consumed [23], eventually leading to death of the animal by cardiac failure [24,25] or other reasons. Evaluating physiological recovery thus becomes a necessary requirement if survival rates want to be addressed in discarded species.

In the southernmost region of Spain, the Strait of Gibraltar, the capture of the Sparidae fish blackspot seabream (*Pagellus bogaraveo*) is an economically important activity. However, the strong currents between the Mediterranean Sea and the Atlantic Ocean [26] have led to the development of a unique fishing technique. As the target species inhabits rocky bottoms below 400 m depth [27], neither bottom trawling nor the usual fishing line and/or longline gears are appropriate in this area. Thus, local fishermen have developed a specific hookline gear called "*voracera*" following the local name to the blackspot seabream, "*voraz*". This gear is highly selective for blackspot seabream [28,29] and, additionally, it allows fish to get on board quickly, being vitally strong and vigorous. Thus, individual monitoring studies were conducted by tagging and releasing seabreams captured by this and similar gears [30,31]. The latter studies mimicked usual discarding activities in the fisheries when captured fish did not reach the minimum commercial size or the total allowable catches (TAC) was exceeded. Discards of blackspot seabream captured by "*voracera*" were mostly represented by fish below the minimum commercial length (33 cm for Atlantic NE and Mediterranean Sea).

The aim of this study was to test the physiological responses to capture in a unique commercial fishery ("*voracera*" hookline gear), and to evaluate survival and recovery of discarded blackspot seabreams. In addition, the physiological recovery of survivors was evaluated, ensuring that discarded fish were not exposed to an extreme situation that could cause a delayed mortality if released into the ocean.

## 2. Results

Blackspot seabream seem especially sensitive to handling in confinement, as in the experiment conducted in ground facilities, six fish died during the first 3 h after their sampling process. It should be mentioned that four fish belonged to unstressed control groups, corresponding to different tanks and sampling times (0 h, 5 h, and 24 h), while two

fish corresponded to stressed groups, but died after samplings at 5 h and 24 h (and thus were considered as recovered animals).

## 2.1. Stress Responses and Physiological Recovery in Ground Facilities

After an acute challenge in the fish husbandry facilities (fish were chased inside the tanks with a hand net for 10 min), blackspot seabream showed a threefold increase in its plasma cortisol levels (Figure 1). After 5 h recovery, cortisol concentrations in the stressed group decreased up to 41% the values described at time 0 h. Though lower, these values were still significantly higher than those of the control group. No statistical differences were observed between both groups 24 h after recovery in water tanks.

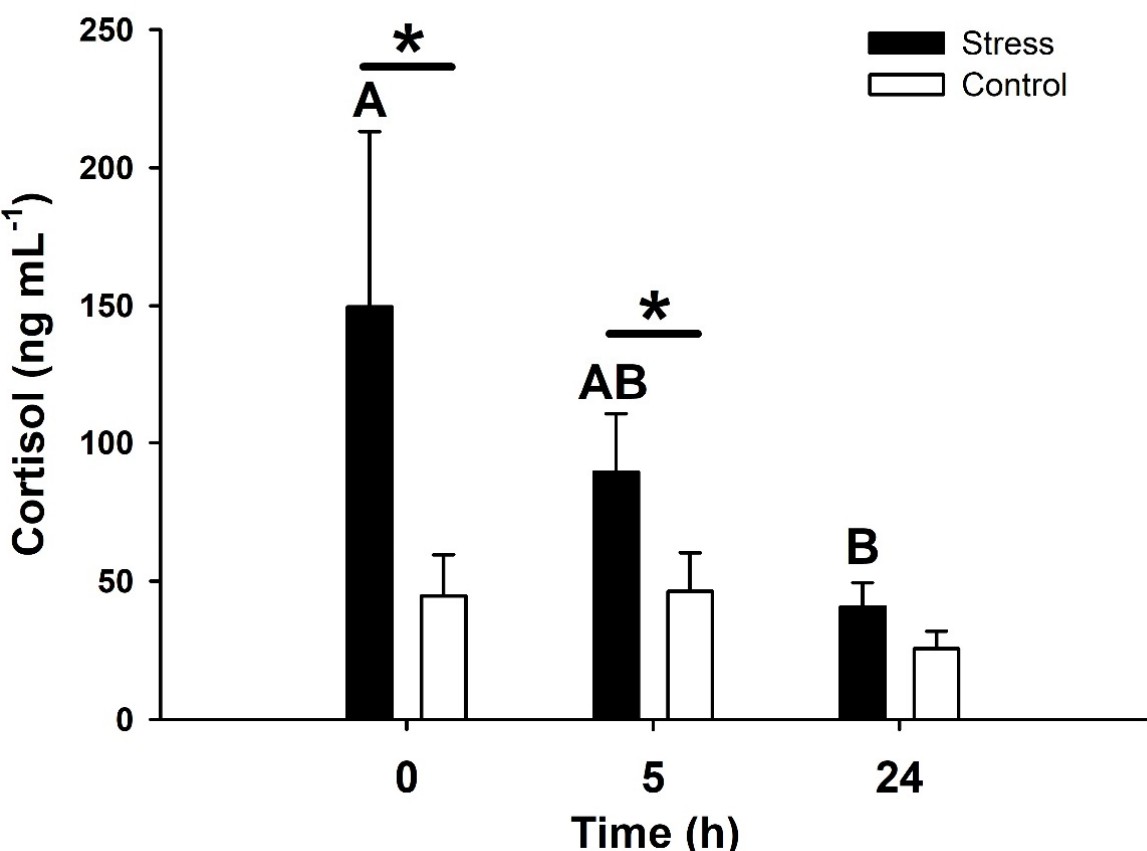

**Figure 1.** Plasma cortisol in blackspot seabream, *P. bogaraveo*, after an acute stress situation in ground facilities. The stressed group (stress, black bars) and the control undisturbed group (control, white bars) were sampled at times 0 h, 5 h, and 24 h after the acute stress challenge. Data are expressed as mean ± SEM (*n* = 9). Different letters indicate significant differences for the stressed group with time. Asterisks (*) indicate significant differences between both groups at each time (*p* < 0.05, two-way ANOVA followed by a Tukey's post hoc test).

A similar response was shown in plasma lactate after the acute stress situation (Figure 2). This metabolite increased its plasma concentrations from circa 1 mmol $L^{-1}$ (control group) to more than 2.5 mmol $L^{-1}$ at time 0 h after the challenge. A gradual recovery of its plasma levels occurred afterwards, with no differences between the stressed and the control groups 5 h later. Plasma glucose in fish maintained in the ground facilities during the time course after an acute stress situation is shown in Figure 2. The control group showed plasma glucose levels with no variations along the experimental time. However, in the stressed group, the concentration of this energy metabolite was significantly enhanced 5 h after the challenge.

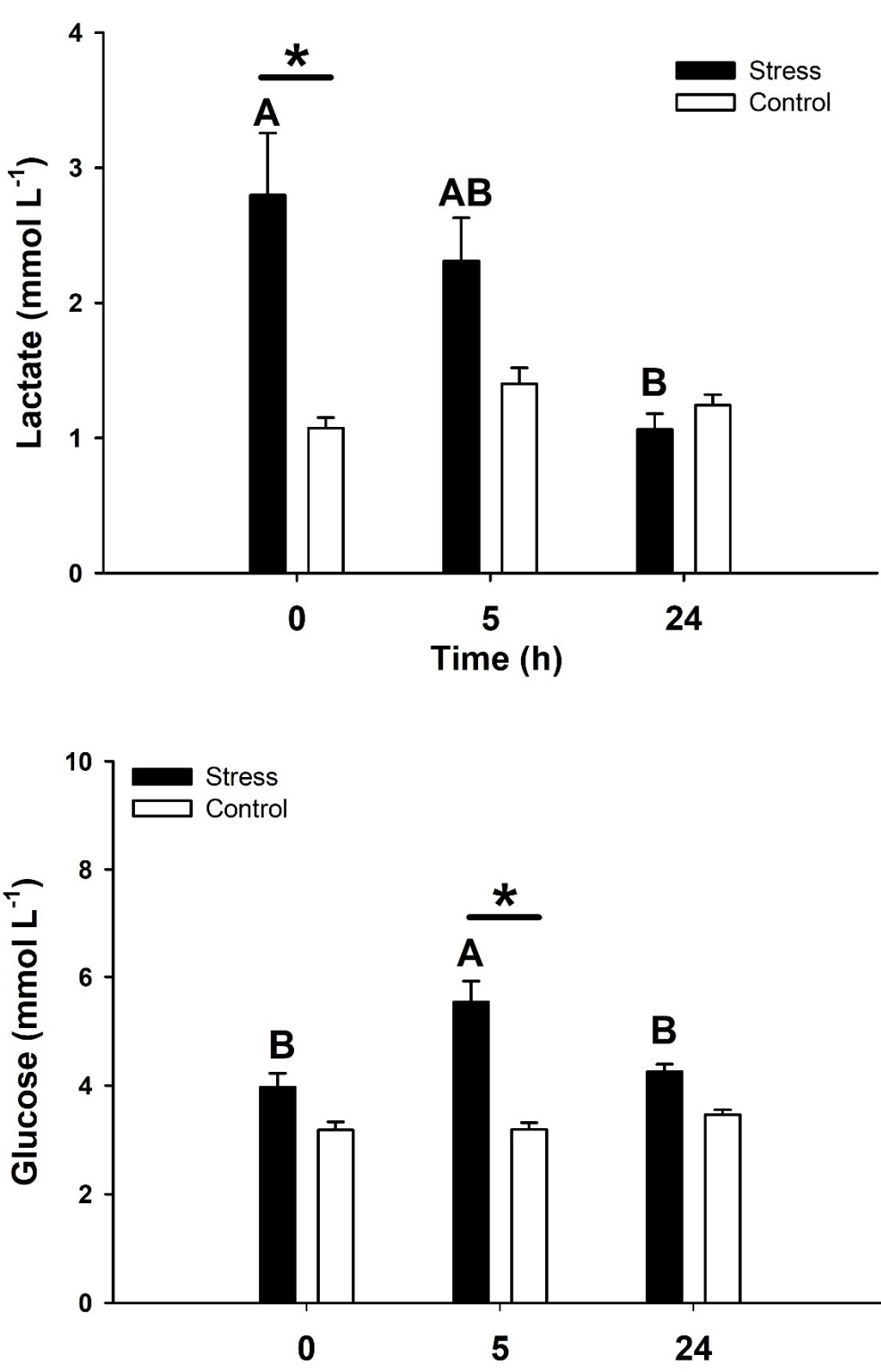

**Figure 2.** Plasma lactate and glucose in blackspot seabream, *P. bogaraveo*, after an acute stress situation in ground facilities. The stressed group (stress, black bars) and the control undisturbed group (control, white bars) were sampled at times 0 h, 5 h, and 24 h after the acute stress challenge. Data are expressed as mean ± SEM (*n* = 9). Different letters indicate significant differences for the stressed group with time. Asterisks (*) indicate significant differences between both groups at each time (*p* < 0.05, two-way ANOVA followed by a Tukey's post hoc test).

Other energy metabolites, such as plasma triglycerides (TAG) and proteins, as well as plasma osmolality, are shown in Table 1. No differences in TAG nor proteins were described for the groups tested. However, plasma osmolality increased significantly in fish immediately after the acute stress process, returning to basal levels after 5 h.

**Table 1.** Plasma triglycerides (TAG), total proteins, and osmolality in blackspot seabream, *P. bogaraveo*, after an acute stress situation in ground facilities. The control undisturbed group (control) and the stressed group (stress) were sampled at times 0 h, 5 h, and 24 h after the acute stress challenge. Data are expressed as mean $\pm$ SEM (*n* = 9). Different letters indicate significant differences for the stressed group with time. The asterisk (*) indicate significant differences between both groups at each time ($p < 0.05$, two-way ANOVA followed by a Tukey's post hoc test).

| Parameter | Group | 0 h | 5 h | 24 h |
|---|---|---|---|---|
| TAG (mmol L$^{-1}$) | Control | $0.75 \pm 0.04$ | $1.01 \pm 0.16$ | $0.74 \pm 0.12$ |
| | Stress | $0.79 \pm 0.09$ | $0.64 \pm 0.08$ | $0.65 \pm 0.07$ |
| Proteins (mg dL$^{-1}$) | Control | $27.5 \pm 1.4$ | $25.7 \pm 0.8$ | $27.5 \pm 1.3$ |
| | Stress | $28.1 \pm 1.0$ | $23.6 \pm 0.8$ | $24.6 \pm 1.5$ |
| Osmolality (mOsm kg$^{-1}$) | Control | $264 \pm 4$ * | $267 \pm 1$ | $263 \pm 1$ |
| | Stress | $291 \pm 3$ [A] | $260 \pm 2$ [B] | $257 \pm 2$ [B] |

*2.2. Survival Rates Onboard*

It should be noticed that none of the captured wild blackspot seabreams floated in the surface when released into the recovery tanks, evidencing swim bladder balance after capture. Moreover, once the fish entered into the tanks, they went to the bottom and actively swam around, exploring the water volume (personal observation). Survival rates onboard a commercial vessel were calculated for both experiences. As there were no significant differences ($p > 0.5$, paired Student's *t*-test) in mortality between the fish employed for the evaluation of survival (fish that were introduced into the recovery tanks immediately after capture, without blood collection), and those employed for the evaluation of the physiological recovery (blood was collected from them) for each fishing set, we assumed duplicated samplings for each fishing set and survival for each fishing set was calculated as the mean of the two experiences. Thus, survival rates 5 h after recovery in water tanks onboard were $90.6 \pm 6.2\%$ (calculated as the mean $\pm$ SEM of all fishing sets).

*2.3. Stress Responses and Physiological Signs of Recovery Onboard*

Physiological recovery in the onboard experience was evaluated. There were significant differences for plasma cortisol, with a significant 33% reduction between fish at 0 h and at 5 h after the challenge (Figure 3).

Energy metabolites, such as plasma lactate and glucose, are shown in Figure 4. Higher concentrations are described for lactate at 0 h, which were significantly different than the values at 5 h. Plasma glucose elicited a significant increase in fish maintained in onboard water tanks for 5 h after being captured compared to fish at 0 h.

As described for the acute stress experiment conducted in ground facilities, plasma TAG did not show any differences in seabreams immediately after capture (0 h) or 5 h after their recovery (Table 2). However, plasma proteins were significantly lowered in those fish 5 h after recovery in onboard tanks. Plasma osmolality was in accordance with the experiment conducted in the fish husbandry, and the highest values are described for those fish immediately after capture (0 h) compared to those values obtained 5 h after recovery.

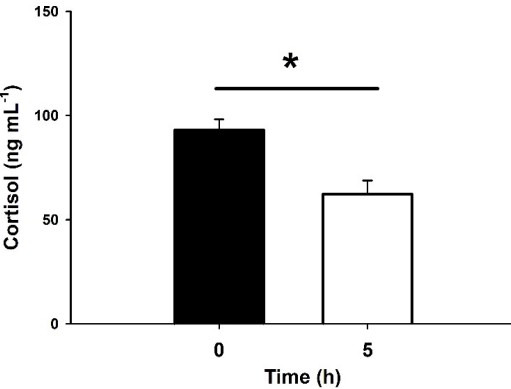

**Figure 3.** Plasma cortisol in blackspot seabream, *P. bogaraveo*, during hookline fisheries in the Strait of Gibraltar (Spain). Fish were sampled immediately after capture (0 h, black bar) and 5 h after their introduction into onboard water tanks (white bar). Data are expressed as mean $\pm$ SEM ($n$ = 33 and 17 fish per sampling time, respectively). Asterisks (*) indicate significant differences between both times ($p < 0.05$, paired Student's *t*-test).

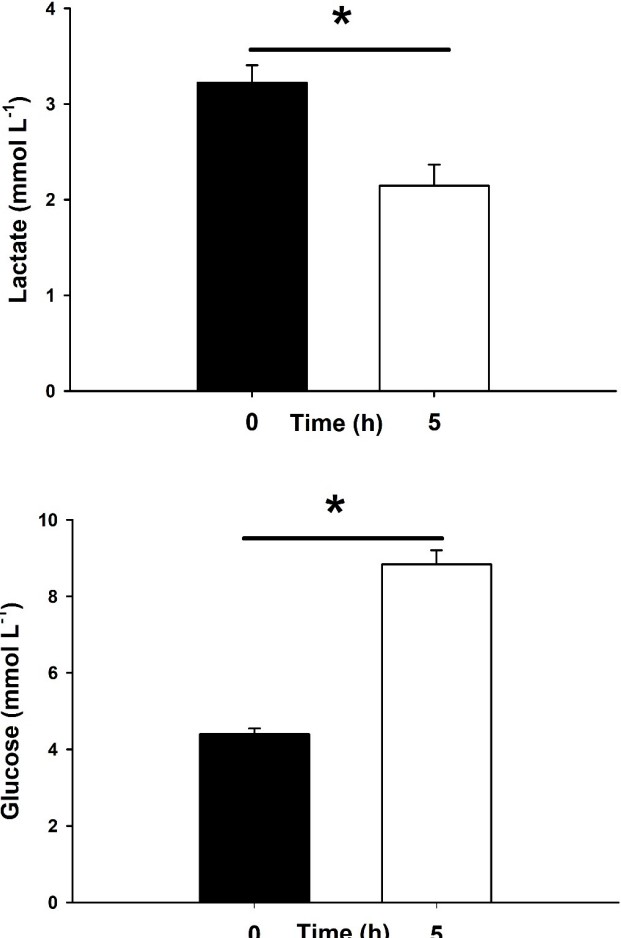

**Figure 4.** Plasma lactate and glucose in blackspot seabream, *P. bogaraveo*, during hookline fisheries in the Strait of Gibraltar (Spain). Fish were sampled immediately after capture (0 h, black bars) and 5 h after their introduction into onboard water tanks (white bars). Data are expressed as mean $\pm$ SEM ($n$ = 40 and 33 fish per sampling time, respectively). Asterisks (*) indicate significant differences between both times ($p < 0.05$, paired Student's *t*-test).

**Table 2.** Plasma triglycerides (TAG), total proteins, and osmolality in blackspot seabream, *P. bogaraveo*, during hookline fisheries in the Strait of Gibraltar (Spain). Fish were sampled immediately after capture (0 h) and 5 h after their introduction into onboard water tanks. Data are expressed as mean $\pm$ SEM (*n* = 40 and 33 fish per sampling time, respectively). Asterisks (*) indicate significant differences between both times (*p* < 0.05, paired Student's *t*-test).

| Parameter | 0 h | 5 h |
|---|---|---|
| TAG (mmol L$^{-1}$) | 1.79 $\pm$ 0.18 | 1.65 $\pm$ 0.18 |
| Proteins (mg dL$^{-1}$) | 26.6 $\pm$ 0.47 | 21.5 $\pm$ 0.4 * |
| Osmolality (mOsm kg$^{-1}$) | 302 $\pm$ 3 | 291 $\pm$ 3 * |

Finally, 1 tagged fish out of 90 was recaptured 10 months after its release. It was recaptured 5 miles away from the original releasing area. This fish increased its body length and weight in 2.0 cm (from 28.0 cm to 30.0 cm total length) and 58 g (from 322 g to 380 g body weight), respectively, during this period.

## 3. Discussion

In the present study, survival rates of a discarded fish species were evaluated through the combined use of classical capture–tag–recapture technique and onboard recovery techniques. In addition, physiological biomarkers were used to ensure the correct recovery of surviving animals. Therefore, physiological responses after an acute stress situation in the blackspot seabream (*Pagellus bogaraveo*) were described. Individuals captured in the Strait of Gibraltar (SW Europe) by the hookline gear called "*voracera*" were acutely stressed, but recovering processes occurred if they were maintained in proper environmental conditions. With the employment of recovery tanks onboard a commercial fishing vessel, this study managed to describe survival rates above 90% in blackspot seabream captured by this gear. European fisheries stakeholders have already taken advantage on these results for this species management in the target fishery of the Strait of Gibraltar, and the first exemption from the discard ban was approved (EU Commission Delegated Regulation 6794/2018).

### 3.1. Physiological Recovery

Increased plasma cortisol levels are described as a primary stress response; thus, blackspot seabream captured by fishing (as mimicked in the ground facilities of this study by chasing) experience a stressful situation. Our results agree with those of other teleost fish such as *S. aurata*, *C. macropomum*, and *S. senegalensis* after an acute stress response [15–17], with the highest cortisol levels during the first hour after the stress, and a decrease afterwards. In those studies, physiological recovery was associated to a sharp decrease in plasma cortisol levels during the first 4 h to 6 h after the challenge, returning to basal pre-stress concentrations in less than 24 h recovery. Fish under chronic stress conditions maintained their plasma cortisol levels significantly elevated for longer periods [32]. Thus, the results obtained onboard a commercial fishing vessel, where blackspot seabream decreased its plasma cortisol levels 5 h after capture and recovery in water tanks, indicated an acute stress response and not a chronic stress situation. In the experiments conducted in this study in ground-based conditions and onboard a commercial fishing vessel, enhanced cortisol similarly dropped between 33% and 41% 5 h after capture, evidencing a recovery process in both experiences. It was previously stated that increased plasma cortisol after capture would incur metabolic imbalances, affecting post-release appetite and suppression of the immune system [4]. These transient effects due to elevated cortisol are recovered in 24 h or less, as seen by the plasma energy metabolites concentrations described in this study.

Cortisol-induced energy mobilization and consumption is evidenced in blackspot seabream by the enhancement of plasma lactate levels after chasing and hookline capture. Plasma lactate is commonly employed as a useful indicator of anaerobic metabolism [4,33], and an excellent secondary biomarker of acute stress responses in fish [15]. Thus, the

increased lactate concentrations in the present study confirmed an acute stress situation in seabream. This increase was also described in the Sparidae species *Pagrus auratus* after simulated angling [34], and in *S. aurata* after air exposure [15]. Lactate increases associated to air exposure/capture highlighted the occurrence of anaerobic glycolysis [4] and glucose mobilization as seen by the increased plasma glucose levels 5 h after capture in the present study. This delayed increase in plasma glucose is explained by the high consuming rates of this metabolite in blackspot seabream during the first hours after the stressful situation, as was described before in *S. senegalensis* [16]. However, notwithstanding the acute stress responses described in the present study, blackspot seabream managed to recover its energy metabolism to basal levels of unstressed fish in less than 24 h after capture. It should be mentioned that the observed differences in the concentration of plasma variables between blackspot seabreams in both studies (wild and aqua-cultured individuals) could be related to the water temperature differences between both experiments (3 °C), amongst other causes. However, it is worth noting that the differences between 0 and 5 h of the stressed groups show similar percentages of change between both experiments, which could indicate that both types of stress (hook fishing and net pursuit) achieve similar physiological responses, allowing us to compare both types of experiments.

Osmoregulatory impairments evidenced by plasma osmolality differences between stressed and control blackspot seabreams were also described for longline-caught cod, *Gadus morhua* [9]. In our study, we described that blackspot seabream managed to recover its osmoregulatory balance within the first 5 h after capture, in accordance to acutely stressed *S. aurata* [18] and *S. senegalensis* [16]. The paralleled increase in both plasma osmolality and cortisol values in teleost fish is a side effect of the mineralocorticoid actions of this hormone [35]. Moreover, cortisol effects may also be related to mobilization of other energy metabolites, such as TAG and proteins [36], unless no differences were described in the experiment performed in ground facilities. The apparent increase in plasma proteins of blackspot seabream immediately after capture (in the onboard experience) could be explained as a side effect of daily rhythms [37] or postprandial effects after eating the bait. However, the fish employed in the land-based experiment were starved for 24 h before sampling, thus avoiding these postprandial plasma imbalances, and the stressed group was sampled in parallel with a control undisturbed group, thus removing putative daily rhythms disturbances.

### 3.2. Survival Rates

In the present study, none of the individuals evidenced buoyancy problems after hookline capture and release into onboard recovery tanks. It was stated in cod that those fish unable to regulate their swim bladder have much higher mortalities than those recovering their air pressure equilibrium [9]. Thus, as physiological recovery after hookline capture was demonstrated for the blackspot seabream in the Strait of Gibraltar, we thus are confident with the survival rates calculated in the present study. The results indicated that 90.6% of captured blackspot seabreams managed to survive under the described circumstances of this specific fishery. We could also hypothesize that survival rates in the wild could even be higher than those calculated herein, although they are similar, the conditions onboard still differ from those in the wild if immediately released after capture. However, other factors such as post-release predation should be also considered [3], which may decrease the calculated survival rates. Our survival results are in accordance with those of *G. morhua* after longline capture [9] and the elasmobranch fish *Scyliorhinus canicula* after bottom trawling [38]. Considering that *S. canicula* is one of the most resilient species to fisheries in the area [39,40], the survival of *P. bogaraveo* after capture by hookline should be noted as a high rate for a hook-captured fish. This high survival rate may be due to the fact that hooking is a quick process, the little damage it does to the animal, and the peculiarities of the blackspot seabream itself. Another relevant aspect that should be considered is the environmental temperature. In this sense, lower water temperatures seem to increase survival rates, which makes it interesting to perform these experiments in

winter and summer in order to get a more comprehensive set of data throughout the year. However, as the averaged seawater temperatures in the surface of the Strait of Gibraltar ranged from 15.8 to 23.3 °C throughout the year, this study was conducted in November when the temperature was 18.3 °C, averaging the yearly temperature in the area. Thus, the survival rates obtained herein could be considered as a good proxy to evaluate survival of captures throughout the year, and slight differences are postulated to exist for this specific fishery. Moreover, in this study, 90 fish were tagged and released alive, and 1 was recaptured 10 months later in the same area. This situation highlights the high resilience or survival capacity of this species captured by the "*voracera*" gear. This was previously demonstrated by the Spanish Institute of Oceanography (IEO), which has been tagging blackspot seabream in the Strait of Gibraltar (aboard vessels from the port of Tarifa, Spain) since the year 2001, and 362 out of 3771 tagged individuals have been recaptured since then [31].

It should be mentioned that mortality rates were oddly high after sampling in the experiment conducted in ground facilities, especially since these animals have been bred in captivity and maintained by professionals who continuously monitor their health status. Surprisingly, all deaths occurred in unstressed/control fish and not in stressed animals. It is possible that the physiological stress shift from metabolically costly, escape-driven responses immediately following hooking/sampling, to a subacute regime over longer chasing/hooking durations, facilitating the recovery of physiological homeostasis, as was described before in the Caribbean reef shark (*Carcharhinus perezi*) after longline capture [41]. An explanation for this mortality could be assumed by the high basal levels of plasma cortisol described in the blackspot seabream compared to other teleost fish [23,42–46]. However, high plasma cortisol levels were also described in *Dicentrarchus labrax*, a species where a cross-regulation between the release of cortisol and catecholamines from the inter-renal and chromaffin tissues, respectively [24], was demonstrated. High plasma concentrations of these hormones may induce a cardiac failure in healthy and not exhausted fish, as was described before in rats [47]. We thus hypothesize that captured blackspot seabream, with a high energy demand due to its high basal levels of cortisol, under a sudden stressful situation not allowing for the mobilization of the energy reserves or for inducing a hypoxic response, may lead to sudden cardiac death. This is in accordance with the described massive and sudden impedance of blood flow into the heart after an acute stress situation, reducing available oxygen and myocardium no longer receiving the optimum balance of energy substrates [48]. This explanation would also describe the high survival rates of the blackspot seabream captured in the Strait of Gibraltar by the local hookline gear "*voracera*", as fish will struggle for circa 10 min, which is enough time to mobilize its energy resources [15] and fuel cardiac cells.

An interesting arising question related to discards is to know how released fish will behave in the future. It was described that those captured fish that managed to survive if released may experience a selective alteration of their physiological responses [49]. This may be translated as a lowered stress response if captured again (proactive fish with low flexibility to environmental changes), or the appearance of reactive fish (for instance by freezing) to threats such as predators. As reactive fish appear to adjust their behavior to new environmental conditions, long-term selection favoring reactive fish [50] could be a benefit to fast anthropic changes. However, to date there is no information on the physiological responses of captured and released blackspot seabreams in the Strait of Gibraltar. Taking into account that there are ongoing tagging studies for this species, it would shed some light on this relevant aspect to analyze stress-related physiological parameters in recaptured seabreams.

Future research directions should point to the implementation of physiology as a useful tool to evaluate survival of captured and released fish [13], since its usefulness is demonstrated in the present study.

## 4. Materials and Methods

### 4.1. Ethics Statement

This study was performed in the husbandry fish facilities of the Spanish Institute of Oceanography (IEO) in Vigo, Spain (Code REGA ES360570189801), onboard a commercial fishing vessel and in a research laboratory in Spain, in accordance with the Guidelines of the European Union (2010/63/UE) and the Spanish legislation (RD 1201/2005 and law 32/2007) for the use of laboratory animals. According to RD1201/2005, the experimental procedures were reviewed by the UCA's Ethics Committee, which approved them. All the people involved were fully accredited to carry out the procedures with experimental animals. This study did not involve the use of endangered or protected species.

### 4.2. Time-Course Responses after Acute Stress in Ground Facilities

In order to evaluate basal homeostatic levels of plasma parameters and assess physiological recovery after hookline capture, an experiment was designed in ground facilities. Laboratory conditions allowed for a finer control of the environmental variables and avoidance of interferences due to on board handling. Blackspot seabream adults of 4 years old ($n = 54$, 270.6 ± 6.0 g body weight and 25.3 ± 0.2 cm total length, mean ± SEM) were bred and maintained at the marine aquaculture facilities of the IEO in Vigo (Spain) until the beginning of the experiment. Fish were randomly allocated into 18 tanks (3 fish per tank) of 500 L (with a surface area of 1.0 m$^2$ and covered by shadowing mesh) in a flow-through system with seawater (35.1 ± 0.1 psu) under a natural photoperiod (November; latitude 42.2328 N) and temperature (ambient temperature of 15.5 ± 0.5 °C) and acclimated for 15 days. Fish were fed once a day with commercial pellets during the experience and were fasted 24 h before sampling in order to avoid physiological disturbances as described for other Sparidae species [42].

The commercial fishing procedures of the hookline employed in the Strait of Gibraltar induce an acute stress situation in blackspot seabreams lasting about 10 min, starting at the moment when the fish bites the hook until it is released back into the water (as described below in Section 4.3). To mimic such an acute stress situation, previous studies challenged fish to chasing inside their tanks [34,51], allowing for the evaluation of primary and secondary stress responses and the physiological recovery processes. Thus, nine tanks were selected as undisturbed controls, and blackspot seabreams from the other nine tanks were chased inside the tanks with a hand net for 10 min (emulating the time that blackspot seabream can spend, at most, fighting on the hook in the fishery called "*voracera*"). Sampling times were settled at times 0 h, 5 h, and 24 h after the acute stress challenge. Three tanks per experimental condition and time were employed. Sampling times at 5 h and 24 h were selected as previous studies described physiological recovery evidence 4 h after an acute stress challenge in the gilthead seabream, another Sparidae species, and complete physiological recovery after 24 h [18].

Fish were captured by hand nets, immediately anesthetized in 0.05% *v/v* 2-phenoxyethanol (P-1126, Sigma-Aldrich, St. Louis, MI, USA), measured in length and weight, and blood was collected (200 μL) by caudal puncture with heparinized syringes. This anesthetic does not seem to affect the stress-related factor analyzed in this study [15,16,42,43,45]. Fish were released back into their tanks after this process, with all the procedures lasting less than 4 min per tank. Blood was centrifuged at 10,000× *g* for 3 min at 4 °C, and plasma was collected afterwards and kept at −80 °C until further analysis of cortisol, glucose, and lactate.

### 4.3. Geographical Location of the Fisheries, Vessel, and Hookline Characteristics

Fish were collected from 14 hookline sets (between 3 and 5 sets per day) in the fishing grounds of the Strait of Gibraltar (South of Spain, Figure 5), locally called "Piedras malas" (35.9190 N, 5.8048 W) and "Discoteca" (35.9214 N, 5.8452 W). The position at the start and the end of each set was recorded using the global position system (GPS). Captures were performed in November 2017 at depths ranging from 234 to 452 m. This study was

conducted onboard the commercial fishing vessel "Nuevo Gabancho" (plate ID# A1-1-2-08, based in the Port of Tarifa, Cadiz, Spain) with a total length of 13 m, engine power of 66.15 kw, gross register tonnage of 14.29 GRT, and capacity for 5 crew members. The vessel included 4 tanks onboard, of 2000 L each, with a flow-through system of seawater collected at 2 m depth.

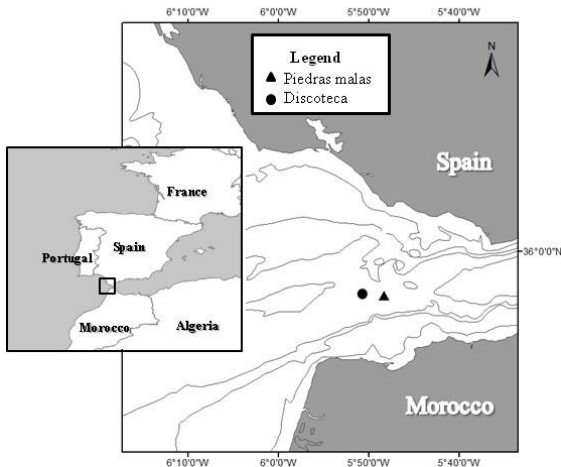

**Figure 5.** Sampled area off the Strait of Gibraltar (the South of Spain). The triangle and the circle indicate "Piedras malas" (35.9190 N, 5.8048 W) and "Discoteca" (35.9214 N, 5.8452 W) fishing areas, respectively.

The fishing set process (Figure 6) starts when the vessel stops, releasing the hookline with baited hooks attached to a concrete weight (15 kg), thus reaching the rocky bottom (Figure 6A). As soon as the weight reaches the seabed the vessel starts moving forward, and a second smaller weight, placed between the hooks and the vessel, puts all the hooks in parallel to the seabed (Figure 6B). While the vessel is in motion, the line breaks just after the first weight, tied with a weaker line to the mother line (Figure 6C), and capture occurs during this process (Figure 6D). After 10 min of capture, the gear is hoisted onboard (Figure 6E). The total time of the fishing process varies between 20 and 35 min.

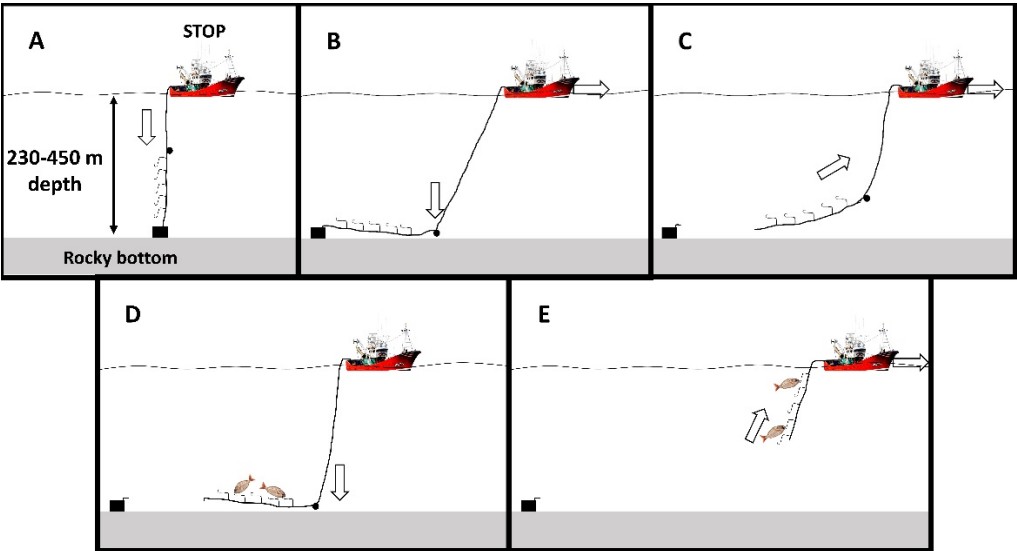

**Figure 6.** Hookline fishing gear, locally called "*voracera*", employed in the Strait of Gibraltar (the South of Spain) for the capture of the blackspot seabream, *P. bogaraveo*. (**A**) Hookline released attached to a concrete weight (15 kg); (**B**) Vessel in motion to place hooks in parallel to the seabed due to a second fishing lead (0.5 kg); (**C**) The main line breaks just after the weight; (**D**) Capture occurs in the bottom; (**E**) The gear is hauled on board.

### 4.4. Survival Rates Onboard

An experiment was conducted to evaluate the survival rates of blackspot seabream onboard. In total, 12 valid sets were conducted, and 106 seabreams were captured (29.4 ± 0.2 cm total length mean ± SEM). The number of sets established for this experiment was >10, with 5 fish per set (in duplicate, since half of the animals were needed to assess physiological recovery), to have an independence of the data that would allow for adequate statistical robustness. The time of gear rolling from the bottom to the surface was 10 min. Once the catches arrived at the fishing deck, 66 fish were employed for the evaluation of the survival rates, measured in length and weight, individually labelled with a rubber band placed in the caudal fin, and introduced into the onboard recovery tanks. All the process lasted less than 30 s per fish, starting from the moment fish were exposed to air, until their release into the tanks. One tank was employed per fishing set at a time, and the total number was 7 ± 2 fish per set (mean ± SEM). Fish were maintained for 5 h in the tanks, as the previous experiment (described in Section 2.2) highlighted that those fish surviving 5 h after the acute stress situation are physiologically recovering their homeostatic levels, while those that did not manage to recover died within the first 3 h after the challenge. Thus, the number of dead animals at the end of this period (5 h) served to calculate the survival rate for each fishing set.

### 4.5. Physiological Recovery Onboard

The physiological effects of hookline capture and the putative recovery response of blackspot seabream were evaluated onboard the commercial fishing vessel. The study was conducted with 40 fish from 8 hookline sets (5 ± 1 fish per set, mean ± SEM) from the total of 106 fish employed to evaluate survival. Captured seabreams were individually labelled with a rubber band placed in the caudal fin, their eyes were covered with a wet tissue in order to improve their welfare, blood was collected as described above (less than 200 µL per fish), and fish were released into recovery tanks onboard. All the procedures onboard lasted less than 15 s per fish after start of air exposure. One tank was employed per fishing set at a time. After 5 h recovery, alive blackspot seabreams were captured by hand nets, covered by a wet cloth, blood was collected again (less than 200 µL per fish), and fish were measured in length and weight before being released back into the ocean, with this process lasting less than 4 min per tank. With this process of repeated blood sampling for each animal, we assume the incorporation of added stress, but we gain statistical power by being able to compare each animal with itself over time. Fish were not anesthetized in this experiment as anesthesia could be fatal to the animals immediately after hookline capture or before being released into the wild. Plasma was obtained after the centrifugation of blood as described above.

### 4.6. Fish Tagging Onboard

All captured animals were tagged immediately before their release into the wild if alive ($n$ = 90). The fish were tagged in the base of the second ray of the dorsal fin by the use of a tagging gun loaded with nylon T-bar anchor tags (Floy tag Inc., Washington, DC, USA).

### 4.7. Plasma Parameters

Plasma variables from both experiments (from the ground-based fish facilities and onboard the commercial fishing vessel) were analyzed. Cortisol was measured using an ELISA commercial kit (Arbor Assay, Ann Arbor, MI, USA). Osmolality was measured with a vapor pressure osmometer (Vapro 5520, Wescor, Logan, UT, USA). Plasma glucose, lactate, and triglycerides levels were measured using commercial kits from Spinreact (Glucose-HK ref. 1001200; Lactate ref. 1001330; Triglycerides ref. 100131101, Spinreact SA, Sant Esteve de Bas, Spain) adapted to 96-well microplates. These methods are based on the phosphorylation of glucose catalyzed by hexokinase (glucose), the oxidation of lactate by lactate oxidase (lactate), and the liberation of glycerol from triglycerides due to lipoproteinlipase followed by conversion to glycerol-3-phosphate by glycerol kinase

(triglycerides). The total plasma protein concentration was determined in diluted plasma samples using a bicinchoninic acid BCA protein assay kit (Pierce, IL, USA, #23225), which is based on a biuret reaction in an alkaline solution, using BSA as a standard. All assays were performed using a Bio-Tek PowerWave 340 Microplate spectrophotometer (Bio-Tek Instruments, Winooski, VT, USA) using KCjunior Data Analysis Software for Microsoft Windows XP.

### 4.8. Statistics

Normality and homogeneity of variances were analyzed using the Shapiro–Wilk test and the Levene test, respectively. Differences between groups in the experiment performed in ground facilities were tested using a nested three-way ANOVA with tank (in triplicate for each group and condition), group (control and stress), and time (0, 5, and 24 h recovery) as factors of variance. In the experiment describing the physiological recovery onboard, differences between groups were evaluated using a repeated measures two-way ANOVA with hookline set and time (repeated measures at 0 h and 5 h) as factors of variance. When necessary, data were logarithmically transformed to fulfill the requirements of ANOVA. When ANOVA yielded significant differences, Tukey's post hoc test was used to identify significantly different groups. As no differences were described in the physiological recovery experiment onboard due to hookline set, a paired Student's *t*-test was employed to evaluate differences between fish at times 0 h and 5 h after recovery in the tanks. Differences between survival in the onboard experiments were evaluated by a Student's *t*-test for dependent samples. Statistical significance was accepted at $p < 0.05$. All the results are given as mean $\pm$ SEM.

### 5. Conclusions

In summary, the present study offers a novel evaluation of survival and (physiological) recovery of captured teleost fish in a previously undocumented fishing gear through a comprehensive set of techniques. As a case study, blackspot seabream (*P. bogaraveo*) captured by the hookline gear called "*voracera*" in the Strait of Gibraltar (SW Atlantic waters of Europe), showed 90.6% survival. This species managed to be physiologically recovered from this process within 24 h, offering relevant data to policymakers who recently approved an exemption for the European discard ban (EU Commission Delegated Regulation 6794/2018). We found that studies onboard commercial fishing vessels should be complemented with studies conducted in controlled ground-based facilities in order to better estimate physiological damages after capture. Furthermore, the methodologies offered herein can be employed for the evaluation of survival and physiological recovery of other organisms, including vertebrate and invertebrate species, captured by other fishing gears.

**Author Contributions:** Conceptualization, I.R.-J., I.S. and J.M.M.; methodology, formal analysis, and data curation, I.R.-J., M.F.-C., I.J.-C., C.B.-M., M.P., E.P., J.G., J.C. and C.F.; writing—original draft preparation, I.R.-J., I.S. and J.M.M.; writing—review and editing, all authors. All authors have read and agreed to the published version of the manuscript.

**Funding:** This research was funded by *Secretaría General de Pesca* (Ministry of Agriculture, Fisheries and Food of Spain).

**Institutional Review Board Statement:** This study was performed in the husbandry fish facilities of the Spanish Institute of Oceanography (IEO) in Vigo, Spain (Code REGA ES360570189801), onboard a commercial fishing vessel, and in a research laboratory in Spain in accordance with the Guidelines of the European Union (2010/63/UE) and the Spanish legislation (RD 1201/2005 and law 32/2007) for the use of laboratory animals. According to RD1201/2005, the experimental procedures were reviewed by the UCA's Ethics Committee, which approved the experiment.

**Data Availability Statement:** The data presented in this study are available on request from the corresponding author.

**Acknowledgments:** The authors want to thank the crew aboard the fishing vessel "Nuevo Gabancho" and M. Suárez for their help during this study.

**Conflicts of Interest:** The authors declare no competing or financial interests. The funding sponsors had no role in the design of the study; in the collection, analyses, or interpretation of data; in the writing of the manuscript, and in the decision to publish the results.

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
