# Peer review of "Survival and Physiological Recovery after Capture by Hookline: The Case Study of the Blackspot Seabream (Pagellus bogaraveo)"

_fishes, doi:10.3390/fishes6040064_

Round 1

Reviewer 1 Report

Manuscript written by Ruiz-Jarabo et al. is well written. I have only small comments to units. I recommend to used mmol/l instead of mM for lactate and glucose plasma concentrations. In table 1, use upper index for letter "A" and "B"

Author Response

Manuscript written by Ruiz-Jarabo et al. is well written. I have only small comments to units. I recommend to used mmol/l instead of mM for lactate and glucose plasma concentrations. In table 1, use upper index for letter "A" and "B".

Answer: Thanks for the comments. All suggested changes have been incorporated into the main text.

Reviewer 2 Report

Dear Editor,

The manuscript entitled "Teleost fish survival and physiological recovery after capture by hookline: the case study of the blackspot seabream (Pagellus bogaraveo)" is very interesting. The study is well structured and the results  supports the conclusions. Anyway the "method", "results" and "discussion" sections need several revisions, before that the manuscript could be  considerated acceptable for the pubblication. Please see the file pdf for the specific comments. Moreover the authors should discuss in more deep way the results about the plasmatic  parameters. So taking into consideration the so interesting results of this study, I encourage the authors to address the comments and submit a revised version of the manuscript.  

Author Response

Answer: All comments have been included in the manuscript except one in which the Reviewer suggests that we incorporate the results of the growth rates of the recaptured animals. This is due to the fact that in the present study there has only been the recapture of a marked individual, so there is not enough statistical information to be of scientific interest. However, for future studies, the information regarding that specific individual has been incorporated into the text. 

Reviewer 3 Report

Ruiz-Jarabo et al. offer a novel test of the physiological stress and survival associated with the voracera fishing method in blackspot seabream. I think that this is a straightforward fishing stress physiology study that overstates its novelty in some instances. Further, I feel that the authors could do more to demonstrate whether their field and lab studies are comparable in the stress response they evoke, given that there are several key methodological differences between the stressor techniques (e.g., air exposure, repeated sampling). Overall, the manuscript would benefit from additional clarification, and I feel that all of the suggested changes below can be incorporated by the authors.

Major Comments

I do not think that this study achieves its stated objective of testing “novel methodology to evaluate survival and recovery of discards”. This is a straightforward fishing stress physiology study and does not develop any novel techniques or biomarkers. Instead, I think the novelty lies in testing the physiological responses to capture in a unique commercial fishery (i.e., voracera).

I am somewhat concerned by the mortality that occurred in the lab study, especially in the control group.

The authors should consider testing for differences in measured stress markers at common time points (i.e., 0h and 5h) between the lab and field-based components of their study. This will demonstrate whether their lab-based approach was similarly stressful to fish as capture in the wild and will add value to this study. At present, the trends appear to be similar between the two methods (i.e., lab vs field), but the magnitude of the response appears to be greater in the wild caught fish, suggesting that the stressor used in the lab-based study may not be wholly representative of capture. This may also have been attributed to the difference in temperature (~3 degrees) between the lab and field studies.

Minor Comments

Lines 90-94: Why did fish die after sampling in the control group? Also, why did more fish die in the control group than in stressed group? Could this have to do with the general condition and health of fish used in the study?

Lines 141-142: If survival rates were calculated using fish from both studies, does this mean that blood was collected from fish that eventually died? Were these data included with fish that survived in analyses? The authors should clarify (later, in the methods) how they handled blood data from fish that survived and died.

Lines 181-185: This is an interesting personal observation, but I am not sure it warrants inclusion in this manuscript.

Lines 273-274: Alternatively, this could suggest extremely low survival capacity.

Lines 315-318: Is this part of the manuscript or guidelines from the journal for writing the discussion?

Lines 351-352: Do the authors have data on how long fish could be chased before showing signs of exhaustion? Also, a statement of why fish were not air exposed might be useful, given that air exposure is a common occurrence in fishing scenarios.

Line 406: The sample size in figure 4 is 33-40, but this section states that 36 fish were sampled. Please clarify the sample size of fish used for characterizing physiological recovery on board. It would also be helpful if the authors could present precise numbers for sample sizes in graphs instead of presenting ranges of sample sizes, especially in figures with only 2-3 bars.

Line 408: I do not think “suffering” is the correct word to use in the context of fish. Perhaps use “disturbance” instead?

Line 411: How long was air exposure?

Line 412: Were different fish sampled at 5h or were the same fish sampled both at 0h and five 5h? The way the sentence is written, it implies that individual fish were sampled twice (i.e., “blood was collected again”). This introduces another difference between the lab and field studies. There is reasonable evidence to suggest that repeat sampling of individuals exacerbates the stress response relative to sampling individual fish once across different time points. This needs to be clarified.

Line 451: I think that the authors offer a novel evaluation of a previously undocumented fishing gear, but the methodology the authors used to evaluate stress and survival is not novel. Please change this wording throughout.

Lines 458-460: I think the authors could do more to demonstrate whether their controlled ground-based experiments approximate the stress associated with their onboard fishing vessel study. There are a number of key differences in experimental design between the two techniques. Both techniques evoke a stress response, but how similar is that response between the two techniques?

Author Response

Major Comments

I do not think that this study achieves its stated objective of testing “novel methodology to evaluate survival and recovery of discards”. This is a straightforward fishing stress physiology study and does not develop any novel techniques or biomarkers. Instead, I think the novelty lies in testing the physiological responses to capture in a unique commercial fishery (i.e., voracera).

Answer: The aim of the study has been modified accordingly.

I am somewhat concerned by the mortality that occurred in the lab study, especially in the control group.

Answer: The mortality in the lab study was something unexpected that only affected to un-stressed or recovered fish… We included in the manuscript a brief discussion of what we considered was the main cause of it.

The authors should consider testing for differences in measured stress markers at common time points (i.e., 0h and 5h) between the lab and field-based components of their study. This will demonstrate whether their lab-based approach was similarly stressful to fish as capture in the wild and will add value to this study. At present, the trends appear to be similar between the two methods (i.e., lab vs field), but the magnitude of the response appears to be greater in the wild caught fish, suggesting that the stressor used in the lab-based study may not be wholly representative of capture. This may also have been attributed to the difference in temperature (~3 degrees) between the lab and field studies.

Answer: The comment of the Reviewer is more than welcome and we thus have included some changes to the manuscript to clarify this issue. The major inclusion is found in the Discussion section, and it says: “It should be mentioned that the observed differences in the concentration of plasma variables between blackspot seabreams in both studies (wild and aquacultured individuals) could be related to the water temperature differences between both experiments (3 ºC), amongst other causes. However, it is worth noting that the differences between 0 and 5 hours of the stressed groups show similar percentages of change between both experiments, which could indicate that both types of stress (hook fishing and net pursuit) achieve similar physiological responses, allowing us to compare both types of experiments.”.

Minor Comments

Lines 90-94: Why did fish die after sampling in the control group? Also, why did more fish die in the control group than in stressed group? Could this have to do with the general condition and health of fish used in the study?

Answer: The main reason of this mortality is unknown (although possible explanations have been incorporated into the manuscript), but causes due to the health of the fish can be eliminated, since the individuals used in the experiment on land have been breed in captivity and are subjected to a continuous review of their physical conditions by professionals. This information was also incorporated into the manuscript.

Lines 141-142: If survival rates were calculated using fish from both studies, does this mean that blood was collected from fish that eventually died? Were these data included with fish that survived in analyses? The authors should clarify (later, in the methods) how they handled blood data from fish that survived and died.

Answer: No blood was collected from dead fish. Thus, only survivors were sampled to evaluate physiological recovery. This information is included, as suggested by the Reviewer, in the M&Ms section.

Lines 181-185: This is an interesting personal observation, but I am not sure it warrants inclusion in this manuscript.

Answer: The Reviewer is right and this information has no scientific relevance, since it only comes from an individual, but we consider that in the future someone could make use of it. That is why we prefer to leave it in the manuscript, if the Reviewer allows it.

Lines 273-274: Alternatively, this could suggest extremely low survival capacity.

Answer: According to previous studies of capture-recapture, this information represents a great percentage of recaptured animals. Especially considering that these fish are found in international waters where fishermen may not have reported their captures. We would prefer to maintain this sentence in the final manuscript in case someone in the future can make use of this information.

Lines 315-318: Is this part of the manuscript or guidelines from the journal for writing the discussion?

Answer: This was a sentence that was left uncorrected in the previous version and we apologize for this. Now it can be read as: “Future research directions should point to the implementation of physiology as a useful tool to evaluate survival of captured and released fish, since its usefulness has been demonstrated in the present study”.

Lines 351-352: Do the authors have data on how long fish could be chased before showing signs of exhaustion? Also, a statement of why fish were not air exposed might be useful, given that air exposure is a common occurrence in fishing scenarios.

Answer: We have no information on how long blackspot seabream could be chased before showing signs of exhaustion. We decided to chase the fish instead of exposing them to air, as chasing may emulate the time that blackspot seabream can spend, at most, fighting on the hook in the fishery called “voracera”. Air exposure for 10 minutes does not occur with this fishing technique and may mislead the results. In our onboard experiment, we have calculated a maximum of 30 seconds per fish of air exposure before releasing them into the recovery tanks (or into the ocean once sampled).

Line 406: The sample size in figure 4 is 33-40, but this section states that 36 fish were sampled. Please clarify the sample size of fish used for characterizing physiological recovery on board. It would also be helpful if the authors could present precise numbers for sample sizes in graphs instead of presenting ranges of sample sizes, especially in figures with only 2-3 bars.

Answer: We would like to thank the Reviewer for the sharp analysis. The comment is right and the total number of sampled fish for blood collection was 40 instead of 36. This mistake has been now corrected in the manuscript.

Line 408: I do not think “suffering” is the correct word to use in the context of fish. Perhaps use “disturbance” instead?

Answer: We appreciate the comment. The sentence is now written as “eyes were covered with a wet tissue in order to improve their welfare”.

Line 411: How long was air exposure?

Answer: The maximum recorded time of air exposure was 15 seconds per fish, as described in the text.

Line 412: Were different fish sampled at 5h or were the same fish sampled both at 0h and five 5h? The way the sentence is written, it implies that individual fish were sampled twice (i.e., “blood was collected again”). This introduces another difference between the lab and field studies. There is reasonable evidence to suggest that repeat sampling of individuals exacerbates the stress response relative to sampling individual fish once across different time points. This needs to be clarified.

Answer: All survivors that were previously sampled for blood collection were sampled again 5 hours after recovery in water tanks. Thus, blood was collected twice (less than 200 µL per fish at each time) at times 0 and 5 hours. We also have included this sentence: “With this process of repeated blood sampling for each animal, we are assuming the incorporation of added stress, but we gain statistical power by being able to compare each animal with itself over time”.

Line 451: I think that the authors offer a novel evaluation of a previously undocumented fishing gear, but the methodology the authors used to evaluate stress and survival is not novel. Please change this wording throughout.

Answer: The Conclusions have been modified according to this suggestion.

Lines 458-460: I think the authors could do more to demonstrate whether their controlled ground-based experiments approximate the stress associated with their onboard fishing vessel study. There are a number of key differences in experimental design between the two techniques. Both techniques evoke a stress response, but how similar is that response between the two techniques?

Answer: We have included some changes in the manuscript to accommodate for this suggestion.

Round 2

Reviewer 2 Report

Dear Editor, 

the authors in this revised version of the manuscript effectively addressed all comments and suggestions received.

Reviewer 3 Report

Ruiz-Jarabo et al. have produced a substantially improved manuscript that I think is now suitable for publication.